# Motivational discourse and campaign-style mobilisation: A positive discourse analysis of language poverty alleviation discourse in China

Run Li[1☉], Yating Yu[2☉]*, Tayden Fung Chan[3☉]*

1 Department of Linguistics and English Language, Lancaster University, Lancaster, United Kingdom, 2 The Department of Communication, Faculty of Social Sciences, University of Macau, Macau SAR, China, 3 Institute of Policy Studies, Lingnan University, Hong Kong SAR, China

☉ These authors contributed equally to this work.
* yatingyu20170901@outlook.com (YY); fungchan202108@gmail.com (TFC)

**Data Availability Statement:** Third party data cannot be shared publicly because the database used in this study is only available through purchase from China National Knowledge

## Abstract

Many studies have investigated language poverty related to aspects of economic assistance, water scarcity, gender inequality, climate change, etc. However, a focus on language policy discourse has been lacking. Language policy discourse is vital because it can be deemed as preliminary to language poverty alleviation action, influencing the success of its implementation. To fill this gap, this study employed positive discourse analysis to investigate discursive strategies used in the discourse of language poverty alleviation in Chinese language policies. The finding shows that through four main discursive strategies–nomination, argumentation, perspectivisation, and predication–official documents concerning language poverty alleviation constructed motivational discourse and applied campaign-style mobilisation to encourage people to follow the implementation of language poverty alleviation. This study sheds light on the official purpose of using certain discursive strategies in language poverty alleviation discourse and some ideological implications behind it and sets an example of the popularisation of official languages to tackle poverty alleviation from a linguistic perspective for other countries/regions.

## Introduction

Poverty is a worldwide phenomenon, afflicting developing countries in particular [1–3]. Poverty has historically been regarded as an unavoidable trend because the low production of non-industrialised economics cannot keep up with the rapidly growing population [4], which leads to "famine, starvation, and deprivation of basic necessities" [5]. As a result, poverty alleviation is urgent in order to lift people out of poverty [6]. Poverty alleviation can be seen as a serious action to meet the needs of the poor [7], including productive employment, microfinance, social insurance schemes, social assistance payments, etc. [8]. In addition to economic interventions in poverty alleviation, there are numerous other approaches and perspectives presented to address poverty, such as dealing with gender inequality [9] and climate change [10, 11], as well as water scarcity and unequal water distribution [12].

Infrastructure (CNKI) with subscription. The first article (https://www.cnki.net/KCMS/detail/detail. aspx?dbcode=CPFD&dbname= CPFDLAST2021&filename= SWYS202105004051&uniplatform=OVERSEA&v= oybIDHWR2O6QIMte9g02kG0YQl7nll CIS6JXAsAxO71KYVq51NSgxwk 861Zbhl6DPTjoHubEy_8%3d); the second article (https://www.cnki.net/KCMS/detail/detail.aspx? dbcode=CPFD&dbname= CPFDLAST2021&filename= SWYS202105001005&uniplatform=OVERSEA&v= oybIDHWR2O41X5s58b-xkLcZ9xYWl_ Ow4I9cpMkDsP7L9fbXvAwEywJAU jkePhzGDd10kPx9TY4%3d); and the third article (https://www.cnki.net/KCMS/detail/detail.aspx? dbcode=CPFD&dbname= CPFDLAST2021&filename= SWYS202105002008&uniplatform=OVERSEA&v= oybIDHWR2O5bUFDW5pAR_keqx0WgTgFp5- OGvx7J21bxUs6ocNge_ QNE4Y3ylcUbfeHUA98iXo4%3d) linked here were used in this study. Others would be able to access or request these data from China National Knowledge Infrastructure (CNKI) in the same manner as the authors. The authors did not have any special access or request privileges to the data from China National Knowledge Infrastructure (CNKI) that others would not have.

**Funding:** The authors received no specific funding for this work.

**Competing interests:** The authors have declared that no competing interests exist.

Language as an essential component in literacy education should not be overlooked and is a critical factor in poverty alleviation for several reasons [13, 14]. Firstly, language skills are essential for finding employment. Without these skills, people may be limited to low-paying, low-skill jobs that offer little opportunity for advancement. Secondly, language is the primary means of communication. When people can communicate effectively, they can access information, resources, and opportunities that can help them participate in economic activities and lift themselves out of poverty. Thirdly, language skills empower people to make informed decisions about their lives, access healthcare, and make intelligent choices for their finances. Subsequently, language poverty alleviation (LPA) has attracted scholars' attention in recent years [15–18].

LPA is an action that alleviates poverty in specific regions by developing language policy and language planning [18] in order to potentially generate economic capital transferred from linguistic capital [19]. In China, the focus of LPA is to popularise Mandarin, particularly in impoverished areas [17], because the popularisation of Mandarin can promote the process of neoliberalisation associated with free-market capitalism, which further contributes to economic development [20] while also preserving and advocating minority languages [21] to ensure the flourishing of minority culture in China. The role language development plays in poverty alleviation cannot be underestimated.

LPA can be reflected through the use of language features and patterns in a global context. Underscoring the importance of language in alleviating poverty, Abdurraheem and Suraju [22] investigated how linguistic features, such as lexical repetition, lexical collocation, synonyms, presuppositions, and implicature, are constructed in the Quran, the Islamic holy book, thereby reflecting the significance of Zakat in promoting poverty alleviation in Nigeria and showcasing the power and function of linguistic features in particular discourses in inspiring people to take poverty alleviation actions. Zakat is a charity scheme that asks every wealthy Muslim to make a donation at the beginning of a lunar year. In this case, the investigated recipients are likely to be limited to individuals who are familiar with Zakat. Therefore, the effect of LPA that is prompted by linguistic features in the Quran might be low. As a result, Abdurraheem and Suraju's study [22] does not take into account how LPA affects a larger population in society.

In addition to linguistic features, indigenous/minority languages also serve a critical role in poverty alleviation. For instance, Igwebuike [23] conducted an LPA study in West Africa. Through the lens of the United Nations Millennium Development Goals, he discovered that indigenous West African languages as people's linguistic capital are still important and can help people seek economic abundance despite being excluded due to the use of ex-colonial languages. Similarly, Li et al. [19] investigated the impact of minority languages on poverty alleviation. They discovered that Hani (a minority language in China) could be transferred into economic capital from linguistic capital because multilingual products are indeed favoured by consumers, particularly in tourism. The importance of developing indigenous/minority languages to reduce poverty is highlighted by these studies. That being said, few studies have investigated how an official language can be used to reduce poverty and the mechanisms that underlie its promotion.

Li's [24] study highlighted the popularisation of an official language to tackle poverty alleviation. Li [24] investigated the role of the national lingua franca in poverty alleviation in China through fieldwork in the Tibetan region of Qinghai province. She discovered that Mandarin, serving as an inter-ethnic lingua franca, helps to break down barriers between minority groups, promotes communication and commercial interaction, and furthers economic development. Li's [24] study demonstrates how the promotion of Mandarin can contribute to China's economic development, but the discursive mechanism in popularising Mandarin, which significantly affects the outcomes of its promotion, is not explained.

Campaign-style mobilisation is a crucial method for LPA, which entails implementing policies through an exceptional mobilisation of administrative resources, backed by strong political support. Its aim is to effectively address the issue of decoupling between regulatory enforcement and compliance [25]. The political context, which influences how institutional change takes place, can be altered through campaign-style mobilisation [26]. During the Maoist era (1949–1966) and to a lesser extent during the reform era, the Chinese Communist Party heavily relied on campaign-style mobilisation to enact policies, with the primary objective of reshaping the social and economic structure [27]. Effective campaign tactics can have long-lasting effects by accelerating the pace of political and legal development towards improved governance and regulatory outcomes [25]. The studies on movements related to campaign-style mobilisation vary greatly depending on the context. While Western historians tend to focus more on movements initiated by the underclass or other marginalised groups, aiming to oppose the ruling authorities from the bottom up, the majority of mass movements in China are initiated by the Party and the nation, starting from the top and trickling down [28].

In the Chinese context, township government workers are informed about the top priority task and then engage in campaign-style mobilisation to facilitate the enforcement and achievement of these prioritised tasks [29]. Unlike the mass campaigns of the 1950s to 1970s, where entire populations were mobilised, the reform period campaigns focused on organising local officials and concentrating efforts on specific policy objectives. This approach enabled the effective implementation of policies at the local level [27]. Furthermore, campaign-style mobilisation often initiates national movements, in which the elite acts as a master, guiding a group of individuals as they contend with other groups that share similar characteristics [28].

While some studies have redirected their focus towards LPA concerns within both domestic and international contexts, there remains a limited body of research that delves into the construction of Chinese LPA discourse, especially in the context of campaign-style mobilisation. The investigation in this field is critical because we can detect and evaluate how the Chinese government formulates an LPA motivational discourse to inspire people and encourages them to support certain language policies. The working definition of motivational discourse in this study is certain types of spoken or written language in materials (e.g., advertisements, business reports, and official documents) that can motivate and mobilise people to make positive social changes in their lives by employing particular discursive strategies. This type of discourse aims to encourage and inspire the audience/reader to take action to achieve a goal that seems unachievable; it can also further particular political or ideological goals and potentially facilitate serious social changes. In this study, the official documents of Chinese language policies regarding LPA are the data under investigation. These materials contain discursive strategies designed to motivate Chinese people to embrace the LPA plan. According to Li et al. [19], the popularisation of Mandarin is emphasised by the Chinese government, and the importance of some minority languages does not receive equal attention. In such a context, how discourse in language policy in LPA convinces and motivates people to support the popularisation of Mandarin is a question that needs to be investigated.

To fill this gap, this study proposes three research questions below to investigate discursive strategies employed in official documents regarding LPA in China from the perspective of positive discourse analysis (PDA), and to investigate how these strategies work.

1. What discursive strategies does the government use in language policies regarding LPA?

2. How do these discursive strategies formulate a motivational discourse to accelerate the implementation of LPA policies?

3. What are the ideological implications for using certain discursive strategies?

## Data collection

This study used purposive sampling, which considers the whole data set of interest, and generalisations are limited to the policy documents analysed. Since the aim of this study is to explore how the official discourse of LAP was formed to motivate people to follow the LPA initiative before the official actions of LAP are implemented, the criteria for article selection are as follows. First, selected articles should represent the official stance. Second, selected articles should serve as preliminary guides to LPA actions. The selected articles have special significance and are "highly valued in the community or [. . .] [have] special significance in some domain such as history or politics [and, thus, can be treated] as artefacts–objects of study in their own right" [30]. Additionally, exceptional single texts can inspire courageous deeds, change the course of history, and give hope in sad times.

The year 2020 was the capstone year for China in terms of poverty alleviation because the government proposed "to eliminate absolute poverty and to ensure the entry of all impoverished areas into a well-off society by 2020" [31]. In order to reflect and reinforce the contributions of poverty alleviation on a socially full-fledged level, LPA is seen as a vital topic. LPA did not officially receive adequate attention until it was first discussed in *Study of Language Policies in China* (中国语言政策研究报告), an official language policy document published in 2021 [32]. In addition, two other official reports regarding LPA, inextricably entangled with the language policies, are also presented in *Language Development in China* (中国语言文字事业发展报告) and *Language Situation in China* (语言生活状况报告). They show LPA was regarded as a significant subject in the official documents of Chinese language policies in 2021. Although the official documents of Chinese language policies in 2022 have been updated with information regarding language and social developments in China, such as language issues related to the Guangdong-Hong Kong-Macao Greater Bay Area and the improvement of national discourse competence in the global context, LPA is not mentioned at all. Language policies are always formulated depending on social development [33].

1. Based on the criteria for article selection, three articles in language policies documents published in 2021 are suitable for analysis to serve the purpose of this study and were gathered from China National Knowledge Infrastructure (CNKI). CNKI (https://www.cnki.net) offers comprehensive academic literature, including books, journals, and conference proceedings. The articles are written in Chinese. In order to precisely present analyses and findings, literal translation, which is faithful to the original texts, is used to avoid potential interference with translation. A translator used the back translation technique to double-check the accuracy of the translation. They are all based on official documents published by the State Language Commission and are available to the general public in all national libraries of China (Table 1). The State Language Commission is the official institution of

**Table 1. Information of the selected data.**

| Articles | Source | Word count |
|---|---|---|
| 语言扶贫<br>*(Language Poverty Alleviation)* | 语言政策研究报告<br>*(Study of Language Policies in China: 2021)* | 8,155 |
| 推普助力脱贫攻坚<br>*(Mandarin Promotion Facilitates Poverty Alleviation)* | 中国语言文字事业发展报告<br>*(Language Development in China: 2021)* | 8,955 |
| 脱贫攻坚收官年的语言扶贫<br>*(Language Poverty Alleviation in the Year of Eradicating Poverty)* | 语言生活状况报告<br>*(Language Situation in China: 2021)* | 4,237 |

language reform in China, whose responsibilities are enacting language policies, planning medium- and long-term language work, etc.

## Analytical framework and procedure

Martin (2004) first proposed positive discourse analysis (PDA), which "functions to make the world a better place" [34]. Unlike critical discourse analysis (CDA), which deconstructs discrimination based on class, ethnicity, gender, and so on as a result of ideology [34], PDA serves as a complementary perspective to CDA, aiming to identify discourse that has the momentum to promote social development and changes [35]. Hughes [36] describes the function of PDA as "enhancing human flourishing and mitigating social ills", which perfectly echoes the topic of this study and allows us to learn more about how the Chinese government's discourse on LPA inspires people to advocate language policies of learning Mandarin to contribute to poverty alleviation. In other words, rather than completely deconstructing social practices, PDA tends to construct them [34, 37–39].

However, as a newly developed branch, PDA has a limited interrelationship with CDA [35], and the lack of a contextual framework remains the critical weakness of PDA [37]. As a result, from the standpoint of PDA, the authors analysed the data using discursive strategy identification to examine discursive strategies in the LPA discourse because they can be used by the government to construct inspiring information that has the potential to turn social anxiety into optimism [39].

The analysis of this study employs Reisigl and Wodak's [40] discursive strategies (see Fig 1). Discursive strategies are specific practices used to achieve a goal or represent a viewpoint/ideology [23, 39, 40], and linguistic features should be prioritised when analysing discursive

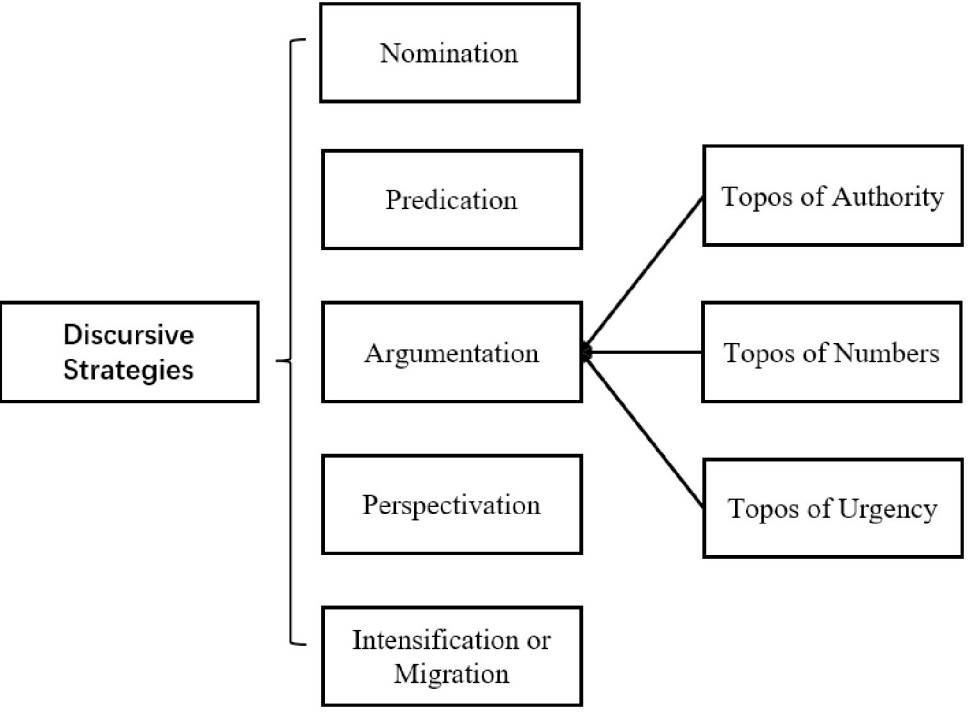

**Fig 1. Analytical framework.**

strategies [39]. In this study, discursive strategies refer to the linguistic features used by the Chinese government in official documents regarding LPA, which have the potential to persuade people of the usefulness of LPA language policies and motivate people to learn Mandarin in order to contribute to poverty alleviation in China. Reisigl and Wodak's discursive strategies are suitable for the analysis of this study because they make an effort to take into account existing knowledge of historical sources as well as the social and political environment in which discursive "events" are situated. Nomination, predication, argumentation, perspectivisation, and intensification/mitigation are the discursive strategies proposed by Wodak [41], and they suffice for the analysis of this study.

*Nomination* aims to construct social actors or events discursively, whereas *predication* aims to depict actions or qualifications of them, such as stereotypical/evaluative attributions or traits [41]. *Argumentation* can justify or refute true or false assertions by employing various types of topoi. Here topoi of authority (highlighting someone in a position of authority), numbers (providing adequate data/statistics), and urgency (conveying a sense of urgency) are found to be prominent in the data. *Perspectivisation* represents a point of view through direct or indirect statements, metaphors, and other techniques, with the goal of demonstrating engagement or detachment. Adverbs, hyperbole, and other language elements are used to modify the illocutionary force of utterances as *intensification* or *migration* [41].

We analysed the data through NVivo 12 Pro software because it allows analysts to develop different coding schemes and do inter-rater reliability tests [42]. The discursive strategies proposed by Wodak [41] were discussed by two analysts as the analytical framework of the study, after which they analysed the data in the first document (39% of the data). Upon finishing the pilot data analysis, the measurement of inter-rater reliability was applied through the Kappa coefficient test, which was calculated as 0.88, indicating a high level of consensus [43]. The coding standard was further discussed by three analysts to eliminate the disagreement of a few instances of divergent views, and the first author then went on to code the rest of the data based on the discussed standard (Fig 2).

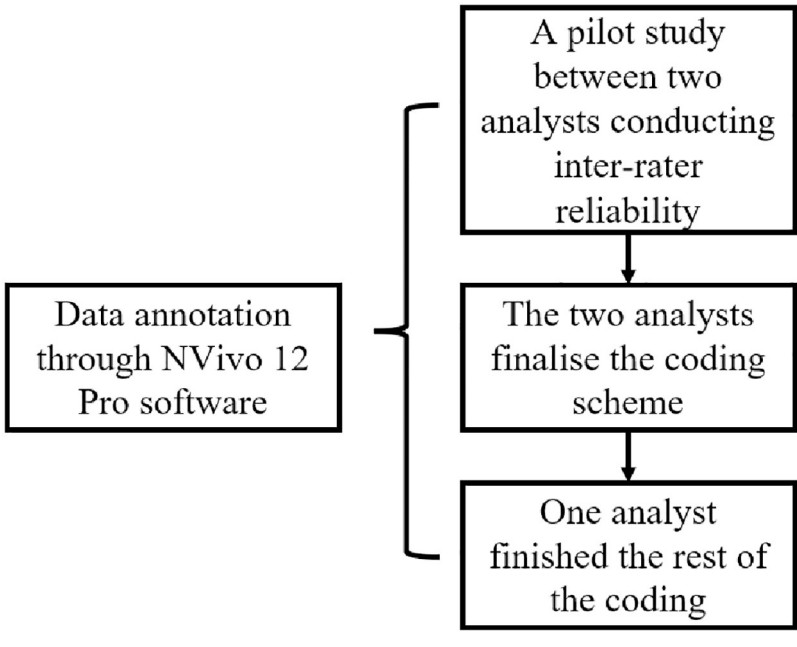

**Fig 2. Analytical procedure.**

**Table 2. Distribution of each discursive strategy contributing to the Chinese LPA discourse.**

| Discursive strategies | Number of instances/percentages |
| --- | --- |
| Nomination | 58 (33%) |
| Argumentation | 46 (26%) |
| Perspectivisation | 38 (21%) |
| Predication | 31 (18%) |
| Intensification/mitigation | 4 (2%) |
| *Total* | 177 (100%) |

## Results

We find that in China, the official LPA discourse is primarily realised through four discursive strategies: referential, predication, argumentation, and perspectivisation. The result is presented in Table 2. The highest proportion of the use is nomination, and the least proportion is intensification/mitigation. The following analysis will mainly focus on the four dominant strategies employed in the LPA discourse.

### Nomination

The primary discursive strategy employed by the Chinese government is the use of nomination in describing the action of poverty alleviation as a battle that encourages people with a common interest to fight poverty together and thereby instils a sense of collectivism in people. Nomination is reflected in two ways in the LPA discourse. First, there is a use of metaphor. Poverty is described as a battle throughout the documents, which may foster people's strong spirit to fight poverty collaboratively [39]. Second, some words or phrases related to the working field help to build a highly efficient and professional image of the government in order to gain people's trust.

1. 各省市县地方部门也都出台了相关工作方案, 为打赢推普助力脱贫攻坚提供落实保障。

   (Local departments in all provinces, cities and counties have also issued relevant work plans to guarantee the ***triumph*** of poverty alleviation through Mandarin promotion.)

2. 以坚定的决心和信心决战决胜脱贫攻坚, 为助力打赢脱贫攻坚战、全面建成小康社会奠定良好基础。

   (***Win the battle*** against poverty with firm determination and confidence to lay a good foundation for helping ***win the battle*** against poverty and build a well-off society.)

In Examples 1 and 2, "打赢 (triumph)" and "打赢攻坚战 (win the battle)" demonstrate that poverty is a common enemy that people must all work together to combat. The metaphor vehicle of *WAR* portrays people as active actors, endowing those involved in the "battle" with hero-like qualities that deserve to be praised. People, particularly those who lack confidence or are hopelessly trapped in a poor situation, are encouraged to participate in the battle against poverty with determination to win the battle. Furthermore, Example 1 indicates that "相关工作方案 (relevant work plans)" issued by "各省市县地方部门 (Local departments in all provinces, cities and counties)" guarantee winning the battle. It suggests that the government always takes the lead and bears great responsibility to ensure the victory of the battle. The Chinese leaders in a great war are always featured with "high competence, moral strength, and the ability to inspire and lead men" [44], thereby constructing a positive (strategic and responsible) image of the government and enabling people to have faith and confidence in the

government's solid support. The use of *WAR* metaphors in this context exposes the campaign-style mobilisation, as the terminology associated with war is intricately linked to the concept of a campaign, as per its etymological roots [45]. This intertwining of war and campaign language creates an atmosphere reminiscent of a campaign, highlighting the vital and pressing nature of the LPA. In the context of Examples 1 and 2, the prerequisite for victory is the promotion and popularisation of Mandarin, which is depicted as a good way to combat poverty. People are more likely to follow the LPA policy because the wartime rhetoric can raise people's morale in the battle and strengthen the unity of people who share common interests [35, 39]. Furthermore, there are implications of collectivism because wartime rhetoric implies that collective efforts and participation are critical in winning the battle. The warfare-like metaphors in the official documents create an urgent tone in which the outcomes of the campaigns imply the destiny of the whole nation.

Linguistic examples from the context of the professional working world can also be found in the LPA motivational discourse. Examples 3 to 5 collectively introduce some specific approaches to implementing LPA.

3. 充分发挥学科优势, 大规模、长时间、高质量地开展线上示范培训。
   (Give full play to the advantages of disciplines, and conduct large-scale, long-term and high-quality online **demonstration training**.)

4. 制定指标考核体系, 强化对推普助力脱贫攻坚工作的指导和督查。
   (Formulate an **indicator assessment system** to strengthen the **guidance** and **supervision** of poverty alleviation by promoting Mandarin.)

5. 专家与基层扶贫人员结合, 专家在评价中发挥主体作用。
   (**Experts** (should) work with grassroots officers who are responsible for poverty alleviation, and **experts** (should) play the main role in the evaluation.)

For instance, Example 3 highlights that the central government does more than just sending commands. It organises "示范培训 (demonstration training)", which is to showcase and explain how to train people (including teachers, officers, etc.) who can teach Mandarin to the poor. The demonstration training is used with relevant departments in various regions to ensure the efficient execution of LPA, revealing the high standard that the government expects from the quality of Mandarin teaching (to the poor). Similarly, "指标考核系统 (indicator assessment system)" in Example 4 motivates relevant officers because their salaries are directly related to their performance, and this systematic approach helps to ensure the successful implementation of LPA [46]. "指标考核系统 (indicator assessment system)", in this study, refers to the guidelines and standards set by the Chinese government for local departments' work on poverty alleviation. The effectiveness of local departments in terms of poverty alleviation can be evaluated using this system. The officials must adhere to the criteria established within such a warlike environment.

Furthermore, "指导和监督 (guidance and supervision)" in a strict way emphasises the great importance attached to LPA, as well as the government's resolve to eradicate poverty through Mandarin promotion. Moreover, the nature of professionalism in LPA is demonstrated by quoting research and expert opinions to gain people's confidence and trust. "专家 (experts)" in Example 5 should not be overlooked. Typically, experts increase the credibility of a specific activity [47], particularly from the perspective of readers who are unfamiliar with LPA in this context, and the professionalism that "experts" bring can win people's trust. Meanwhile, the sense of professionalism persuades people that the implementation of LPA is on the right track and will ultimately be successful. Poverty has a significant impact on all people who are affected by it. Therefore, people in underprivileged communities share common problems

and common interests. The emphasis on professionalism prompts people who share common needs to show more trust in leaders who have expertise in dealing with difficulties [48]. The systematic and standardised preparation work and actions imply campaign-style implementations, which also should involve scientific and large-scale arrangements, ensuring the highly effective execution of specific political agendas and tasks.

## Argumentation

Another discursive strategy found in the official documents is the use of argumentation for the purpose of justifying the LPA policies and thus creating a positive image of the Chinese government. The justification of poverty alleviation through promoting Mandarin can be roughly classified into three aspects: urgency, professionalism, and benefits.

6. 2018年以来, 学界在探讨贫困地区推普方略的同时, 深入研究语言与贫困的基础理论问题和重大实践问题, 推动"语言扶贫"成为近年来语言政策研究的重要话题。
(Since 2018, while discussing the strategy of promoting Mandarin in poverty-stricken areas, academic circles have conducted in-depth research on the basic theoretical issues and major practical issues of language and poverty, and promoted "language poverty alleviation" to become an important topic in language policy research in recent years.)

7. 2018–2019年间, 我国学者以推普助力脱贫实践为核心, 展开了集中的语言扶贫理论研究。
(From 2018 to 2019, Chinese scholars carried out intensive theoretical research on language poverty alleviation with the practice of promoting Mandarin to alleviate poverty as the core task.)

8. 饶高琦、魏晖从语言资源观角度, 将语言扶贫界定为"利用语言资源相对优势获取个体或区域语言能力, 进而增加个体收入或促进区域经济社会发展的一种扶贫方式"。
(From the perspective of language resources, Rao Gaoqi and Wei Hui define language poverty alleviation as "a poverty alleviation method that uses the comparative advantage of language resources to obtain individual or regional language ability, thereby increasing individual income or promoting regional economic and social development".)

9. 会普通话的贫困户2018年的打工收入比不会普通话的贫困户高32.8%; 按照国家3747元农村家庭人均年收入的精准扶贫标准统计, 会普通话家庭的脱贫率增加了20%左右。
(In 2018, the income of poor households who can speak Mandarin was 32.8% higher than that of poor households who do not speak Mandarin. According to the country's targeted poverty alleviation standard of 3,747 yuan per capita annual income for rural households, the poverty alleviation rate of households who speak Mandarin has increased by about 20%.)

Example 6 is a topos of urgency that treats LPA as "基础理论问题和重大实践问题 (basic theoretical issues and major practical issues)", and "重要话题 (an important topic)" particularly "近年来 (in recent years)". The importance placed on LPA, particularly in official political discourse, has the potential to raise people's awareness, making them more likely to engage in this issue, particularly among the poor. "Chinese scholars (我国学者)" in Example 7 demonstrate the topos of authority. They have conducted "集中的理论研究 (intensive theoretical research)" on LPA in recent years, which adds to the point of worthiness and importance. The identity of the scholar, combined with the extensive research they have conducted, increases the credibility of the successful implementation of LPA. Furthermore, scholars regard LPA "为核心 (as the core)", and place a high value on it. This raises people's awareness of the significance of the LPA issue, which is closely related to their daily lives, echoing the implication

conveyed by Example 6. In Example 8, two experts in language and linguistics, Rao Gaoqi and Wei Hui, define LPA as "增加个体收入或促进区域经济社会发展 (increasing individual income or promoting regional economic and social development)". The seemingly authoritative definition explains the advantages of LPA. Scholarly justifications and the presentation of knowledge from numerous specialists also highlight the mobilisation of Chinese campaign-style and elite-led national movements [28]. These movements aim to encourage citizens' participation in the LPA. Example 9 reflects a similar message through the topos of numbers, which are more persuasive, as concluded by empirical studies. The statistical data can show the effectiveness of promoting Mandarin in alleviating poverty and the extent of improved economic status among the designated areas.

As a result, experts' definitions and statistical evidence from fieldwork research solidly demonstrate the existence of benefits–Mandarin promotion can promote poverty alleviation. People will show more support and trust in the government on this issue once they are convinced that LPA can indeed lift them out of poverty and improve their quality of life. Through the augmentation strategy, people and officials can understand the reasons for implementing LPA and its expected outcomes. Moreover, argumentation emphasises the urgency to raise awareness of the importance of carrying out LPA. People may think that if LPA fails to be achieved, individual welfare and social development will be jeopardised. Overall, the argumentation largely enhances the policy legitimacy and persuades people to accept what is promoted by official authorities as being correct.

## Perspectivisation

Apart from argumentation, perspectivisation is also a discursive strategy used to enhance the legitimacy of LPA and publicise the policy message. By directly quoting opinions from the official meeting, documents, and experts, the Chinese government demonstrates the importance and necessity of promoting Mandarin and also shows that some tangible and systemic actions have been taken in this regard. Furthermore, the perspective of protecting minority languages appears to moderate the seemingly arbitrary notion of "Mandarin as the core" and ensure the equity and diversity of language development.

10. 会议把"坚定不移推广普及国家通用语言文字, 助力决战决胜脱贫攻坚"列为专项任务。

(A special task that the conference proposed is that the "national lingua franca should unswervingly be promoted and popularised to contribute to the battle of fighting poverty".)

Setting Mandarin as the primary official language to promote poverty alleviation was on the agenda of the National Chinese Language Conference (NCLC; https://asiasociety.org/education/national-chinese-language-conference) in 2020. NCLC offers a prominent forum for exchanging cutting-edge concepts and the greatest practices in Chinese language teaching and learning. The 2020 conference aimed to "推广普及国家通用文字 (Promote and popularise the national lingua franca)" and treat this as "专项任务 (a special task)", indicating that the priority had been given to Mandarin promotion to alleviate poverty (Example 10). This was the most important task at that particular moment, enabling people to understand the importance of learning Mandarin, as was emphasised in the official meeting. Example 10 demonstrates the significance of Mandarin promotion.

11. 5月18日印发《打赢教育脱贫攻坚收官战总攻方案》, 明确要求全面加强各级各类学校国家通用语言文字教育...

(*The Plan for Successfully Combating Poverty through Education* was published on 18 May.

It explicitly demands that the education needed for a national lingua franca in schools of different levels should be attached more importance.)

Apart from the conference mentioned in Example 10, there are other official documents involved in the LPA motivational discourse that are specifically aimed at promoting Mandarin to fight poverty. The release of these official documents can directly demonstrate the importance that the government currently places on LPA. For instance, in Example 11, the official document The Plan for Successfully Combating Poverty through Education details the precise methods used in the field of education to alleviate poverty. The official document demonstrates actions that have been done by the Chinese government in a thorough and assertive manner [49]. The official document specifically indicates that "国家通用语言文字教育 (education needed for a national lingua franca)" is "明确要求 (explicitly demand)", which can be regarded as a mandatory task for students, particularly for those with limited knowledge of Mandarin. The education is provided in "各级各类学校 (schools of different levels)" to demonstrate its necessity, and it also shows the government's determination to popularise Mandarin in the country. Prescriptive terms such as "unswervingly promote" and "explicitly demand", found in the discourses of official and critical conferences and documents, establish the tone for criteria, direction, objectives, and so on, within the nationally significant LPA movements. These terms effectively function as typical and essential components of campaign-style governance, enabling the successful implementation of the LPA.

12. 王春辉依据语言的性质和功能明确了推普助力脱贫的机理, 并在此基础上构建了推普助力脱贫攻坚效果评估的指标体系。

(According to the essence and function of the language, Wang Chunhui confirms the mechanism of poverty alleviation through promoting Mandarin, based on which he proposes the standard to evaluate poverty alleviation through promoting Mandarin.)

In Example 12, Wang Chunhui, the dean of the Research Centre of Language Governance at Capital Normal University in Beijing, offers an important perspective. "机理 (mechanism)" and "指标 (standard)" show that LPA was well planned and resulted in goals being achieved. Currently, many experts have conducted extensive research on LPA, many of which are also revered in the LPA motivational discourse. In this way, the government is more likely to gain people's trust because the government seeks advice from experts and develops numerous feasible plans rather than simply posting slogans or pressuring people to learn Mandarin without making any effort, as evidenced in the data. Therefore, the importance and necessity of LPA conveyed by the government are critical for arousing positive public sentiment, thus more easily enacting language policies and conducting language planning.

13. 刘金林、马静认为, 兼顾当地少数民族语言的保护和传承, 应是下一步语言扶贫需要考虑和解决的核心问题。

(According to Liu Jinlin and Ma Jing, the protection of and support for minority languages should be the core issue that needs to be considered and resolved in the next step.)

Although the documents set Mandarin as the primary official language to be promoted for the purposes of poverty alleviation, the significance of other minority languages is also emphasised. If Mandarin is solely popularised, people from ethnic minorities are more likely to be alienated. As such, the inclusion of minority languages on the agenda reassures people that their languages and cultures will not be ignored. This practice strengthens people's willingness to collaborate with the government at this critical juncture. Such a strategy of perspectivisation gives the impression that the government has fully considered various factors before making its decision. Furthermore, people with opposing voices who advocate the development of

minority languages would be more willing to compromise. For instance, the experts who did relevant field trips and empirical studies in Example 13 emphasise the importance of developing minority languages. They suggested that "少数民族语言的保护与传承 (the protection of and support for minority languages)" should be "核心问题 (the core issue)" in the future and indicated that, in addition to Mandarin popularisation, the government recognises the importance of promoting minority languages as they have distinct educational functions to cognitively and affectively nurture people, particularly children from ethnic minority backgrounds as evidenced in Example 13 [50]. The promotion of minority languages and Mandarin strengthens minority people's cultural identity [51] and sense of belonging while also accelerating the process of poverty alleviation, which is a desirable scenario for people. It is clear that academic and professional perspectives often feature prominently in documents, lending support to the campaign-style mobilisation of the LPA in China. In a situation like the one presented here, the influence on the public will be even more significant due to the prominence of expert voices and the extensive dissemination of these viewpoints through news articles, reports, and other channels.

## Predication

As for predication, the motivational discourse used in the documents displays the tangible and precise actions that are being and will be taken by relevant departments. Predication expresses the determination to implement the LPA policy. It also conveys the message that the government communicates to the public the ongoing progress in LPA and that the LPA policy will be maintained for an extended period of time to achieve lasting outcomes. It shows the government's determination and resolve.

14. 2020年的相关研究进一步丰富和发展了语言扶贫的内涵, 详细地阐释了语言的减贫效应。

   (The relevant research in 2020 **further enriched and developed** the content of language poverty alleviation and explained in detail the poverty reduction effect of language.)

15. 坚定不移推广普及国家通用语言文字, 助力决战决胜脱贫攻坚。

   (**Unswervingly** promote and popularise the national lingua franca, contributing to the decisive battle against poverty alleviation.)

16. ...省部合作共推、社会多元参与的协同推进体系。

   (**A collaborative promotion system** consists of **provincial cooperation** and **multi-level social participation**)

In Example 14, "语言扶贫内涵 (the content of language poverty alleviation)" is "丰富和发展 (enriched and developed)" by "相关研究 (relevant research)", indicating that LPA is in progress and evolving in a positive direction. "进一步 (further)" modifying "丰富和发展 (enriched and developed)" shows additional improvements have been made to lay the solid foundation for the final success of LPA. The adverb "坚定不移 (unswervingly)" in Example 15 reveals the determination of the government. Despite the many challenges ahead in poverty alleviation due to China's large population [52], the Chinese government does not flinch, and it contributes fully to "脱贫攻坚 (the decisive battle against poverty alleviation)". The unwavering attitude can greatly foster people's collaborative spirit to participate in LPA. As further illustrated in Example 16, "省部合作共推 (provincial cooperation)", "社会多元参与 (multilevel social participation)", and "协同推进 (collaborative promotion)" demonstrate how the government brings together the power and wisdom of all parts of society to engage in LPA, creating a holistic and united impression. The emphasis on collaboration in official documents

may be attributed to the fragmented nature of Chinese local political organisations, where local administrations often operate under different leadership [53]. LPA can lead people to consider that they, too, are a part of society and thus have obligations to contribute to major national issues that are inextricably linked to their welfare, creating a sense of collectivism and unity. Group culture, which promotes the spirit of cooperation and unity, is popular and important in China for instilling a sense of belonging [54]. As a result, fostering collectivism is common in political discourse in China to motivate people. It is worth noting that "省部合作共推 (provincial cooperation)" in the documents elucidating local execution of directives from the central government resonate with the hallmark of campaign-style mobilisation, wherein local officials are coordinated and united to propel a specific cause [27].

## Discussion

The previous sections address and respond to the first two research questions set in this article: 1) The main discursive strategies used by the government to promote LPA include nomination, argumentation, perspectivisation, and predication; 2) Through these four strategies, the motivational discourse is formulated by creating a wartime-like environment, enhancing policy legitimacy with experts' opinions and diverse viewpoints, and providing a foreseeable development path. For the final question regarding ideological implications, the main views are discussed below.

With the use of the four main discursive strategies, several advantages for policy implementation can be highlighted as follows: (1) the government officials are able to put the agenda of fighting poverty as a top priority and allocate resources to achieve the policy outcome; (2) social mobilisation for common goals distracts people's attention from the government's unsatisfactory performance and enhances the authority of local officials; and (3) those who do not cooperate with the government could be morally criticised as spoiling the spirit of collectivism. It should be noted that, as mentioned in the section "nomination", the strategies introduce the scientific implementation and evaluation concept reflected by the training and the "指标考核系统 (indicator assessment system)" [46]. The local officials are required to fulfil the targets set by the upper-level units in order to have better performance appraisals. Thus, LPA provides a good example of how local officials are encouraged to implement the policies set by the central authorities within a warlike environment.

Notably, the Chinese government tends to launch campaign-style mobilisation to implement social policies from time to time [55], and the formulation of the motivational discourse in LPA investigated in this study is a good example. With a pressing atmosphere, implementing social mobilisation as a kind of military preparation is a way to save the cost of management [55]. The institutional setting significantly conforms to the centralised control and requires the lower-level units or individuals to follow the mandates issued forth from the higher-level systems [55]. Such mobilisation with ideological morale is conducive to collaboration within the fragmented structures of the current regime, as mentioned in the section titled "predication". Chinese local political structures are often seen as fragmented due to their diverse leadership hierarchies [53]. The implementation of the LPA holds the potential to enable these fragmented units to effectively collaborate by adhering to a centrally formulated policy. Furthermore, as discussed in the sections titled "nomination" and "argumentation", the campaign-style mobilisation also serves to enhance the legitimacy of the LPA. By creating a sense of urgency and providing logical justifications, the people were convinced to implement the policy [55].

Furthermore, the official line set by the Chinese propaganda authorities usually promotes a pluralistic and inclusive environment [56]. This study shows that even though Mandarin is

designated as a primary language in the measure of alleviating poverty, there is also an intention to protect other minority languages. In the analysis of the perspectivisation strategy, the idea of "Mandarin as the core" appears to be moderated by the viewpoint of minority language protection, which also ensures equity and variety in language development. If Mandarin is the only language that is popularised, people from ethnic minorities are more prone to feel alienated. As such, the fact that minority languages are on the agenda gives people confidence that their languages and traditions will not be disregarded. This perspectivisation strategy makes individuals more inclined to work with the government, lessens the influence of those who are opposed to LPA, and creates the impression that the government fully considers all relevant aspects before making its choice. This practice shows that the mobilisation, conducted in a campaign-style manner, also included subtle strategies to minimise the potential consequences for the preservation of indigenous minority languages.

## Concluding remarks

Through the lens of positive discourse analysis, this study identifies the discursive strategies employed by the Chinese government in LPA and how these strategies formulated the motivational discourse to inspire people to support language policies with the aim of promoting poverty alleviation. This study also addresses the official purpose of using certain discursive strategies in the LPA discourse and selects ideological implications behind it. Last but not least, this study discussed the Chinese government's policy campaign-style mobilisation in relation to the LPA motivational discourse. As mentioned in the introduction, little attention has been given to Chinese LPA discourse construction in the existing literature. This study, therefore, has filled this research gap by investigating the LPA motivational discourse and sheds light on its formulation with different discursive strategies and ideological implications in relation to the campaign-style mobilisation. In this study, we highlight that the LPA motivational discourse serves as a prime illustration of campaign-style mobilisation and sets an example for other countries/regions as to the popularisation of an official language to tackle poverty alleviation from a linguistic perspective. Whether minority languages can be preserved under the current LPA practices is left open for future research.

## Author Contributions

**Conceptualization:** Run Li.

**Data curation:** Run Li.

**Formal analysis:** Run Li, Yating Yu.

**Investigation:** Run Li.

**Methodology:** Run Li, Yating Yu.

**Project administration:** Tayden Fung Chan.

**Supervision:** Yating Yu, Tayden Fung Chan.

**Writing – original draft:** Run Li, Yating Yu, Tayden Fung Chan.

**Writing – review & editing:** Yating Yu, Tayden Fung Chan.

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
