## [Decision Letter · Decision Letter 0]

27 Feb 2023

PONE-D-22-35166A motivational discourse in Chinese language policies: A positive discourse analysis of language poverty alleviation discourse in ChinaPLOS ONE

Dear Dr. Yu,

Thank you for submitting your manuscript to PLOS ONE. After careful consideration, we feel that it has merit but does not fully meet PLOS ONE’s publication criteria as it currently stands. Therefore, we invite you to submit a revised version of the manuscript that addresses the points raised during the review process. a substantial revision is required.Please submit your revised manuscript by Apr 13 2023 11:59PM. If you will need more time than this to complete your revisions, please reply to this message or contact the journal office at plosone@plos.org. Please include the following items when submitting your revised manuscript:A rebuttal letter that responds to each point raised by the academic editor and reviewer(s). You should upload this letter as a separate file labeled 'Response to Reviewers'.A marked-up copy of your manuscript that highlights changes made to the original version. You should upload this as a separate file labeled 'Revised Manuscript with Track Changes'.An unmarked version of your revised paper without tracked changes. You should upload this as a separate file labeled 'Manuscript'.

We look forward to receiving your revised manuscript.

Kind regards,

Bing Xue, Ph.D.

Academic Editor

PLOS ONE

Journal Requirements:

Reviewers' comments:

Reviewer's Responses to Questions

**Comments to the Author**

1. Is the manuscript technically sound, and do the data support the conclusions?

Reviewer #1: Yes

Reviewer #2: Partly

2. Has the statistical analysis been performed appropriately and rigorously? 

Reviewer #1: Yes

Reviewer #2: No

3. Have the authors made all data underlying the findings in their manuscript fully available?

Reviewer #1: No

Reviewer #2: Yes

4. Is the manuscript presented in an intelligible fashion and written in standard English?

Reviewer #1: No

Reviewer #2: No

5. Review Comments to the Author

Reviewer #1: This article presents an interesting study about the discursive strategies employed in the discourse of language poverty alleviation in three Chinese policy-related documents. Overall, the research design is original and technically sound.

A very critical point to note is that the "motivational discourse" seems to be a key concept that the authors put forward via analysis. If so, it would be very necessary to give a working definition of it.

Moreover, perhaps due to the length of the article, the presented examples are individual sentences extracted from the three documents. However, some complicated concepts would be hard to follow without specific contexts (maybe also because they are Chinese specific?), for instance, "indicator assessment system". Therefore, I would suggest the authors add more explanatory texts to elaborate them in their corresponding paragraph before analyzing the strategies. By the same token, I'd appreciate it if the authors could provide the three articles so that I can fully grasp them.

In addition, it would be better if the authors can relate the current study to previous research in the "Discussion" section. Please do consider what this case study may contribute to a broader international academia on discourse studies, policy studies, or other relevant strands.

Please double check wording and grammar. I've highlighted some in the manuscript.

Other comments and minor suggestions were attached in the manuscript.

Reviewer #2: This study employed positive discourse analysis to investigate discursive strategies used in the discourse of language poverty alleviation in Chinese language policies and brought out policy suggestions. However, the argument in the whole article is superficial, and the logic is not perfect enough, so it has not reached the publication standard of the journal.

1. In Introduction, many related researches have been introduced yet in a simple listing way without systematic generalizations or comments concerning each category.

2. The reports mentioned in Data collection should be given the sources.

3. It would be better if the authors can use a diagram to show the analytical framework and procedure of this study.

4. The corpus used in this study has a relatively small scale, with only 177 cases discovered, which is insufficient to draw a conclusion.

5. In every section of strategy analysis, the authors only gave several examples and shortly explained them. Even in Discussion, the analytical work is still not profound enough. However, we hope to see deeper and more theoretical analysis of each strategy, and more in a logical way, e. g. subsuming some patterns.

6. In Concluding remarks, the points of view brought out lack solid background, and it’s hard to tell how the authors reach these conclusions.

7. The English used in this paper can be further polished to reach academic standard.

8. The literatures cited in this paper failed to reflect the cutting-edge researches in this field in recent years.

6. PLOS authors have the option to publish the peer review history of their article (what does this mean?). If published, this will include your full peer review and any attached files.

Reviewer #1: No

Reviewer #2: No

---

## [Author Response · Author response to Decision Letter 0]

6 Jun 2023

Please refer to the the file (rebuttal letter) submitted.

---

## [Decision Letter · Decision Letter 1]

24 Jul 2023

PONE-D-22-35166R1Motivational discourse and campaign-style mobilisation: A positive discourse analysis of language poverty alleviation discourse in ChinaPLOS ONE

Dear Dr. Yu,

Thank you for submitting your manuscript to PLOS ONE. After careful consideration, we feel that it has merit but does not fully meet PLOS ONE’s publication criteria as it currently stands. Therefore, we invite you to submit a revised version of the manuscript that addresses the points raised during the review process.

We look forward to receiving your revised manuscript.

Kind regards,

Bing Xue, Ph.D.

Academic Editor

PLOS ONE

Journal Requirements:

Reviewers' comments:

Reviewer's Responses to Questions

**Comments to the Author**

1. If the authors have adequately addressed your comments raised in a previous round of review and you feel that this manuscript is now acceptable for publication, you may indicate that here to bypass the “Comments to the Author” section, enter your conflict of interest statement in the “Confidential to Editor” section, and submit your "Accept" recommendation.

Reviewer #1: All comments have been addressed

Reviewer #2: (No Response)

2. Is the manuscript technically sound, and do the data support the conclusions?

Reviewer #1: Yes

Reviewer #2: No

3. Has the statistical analysis been performed appropriately and rigorously? 

Reviewer #1: Yes

Reviewer #2: Yes

4. Have the authors made all data underlying the findings in their manuscript fully available?

Reviewer #1: Yes

Reviewer #2: Yes

5. Is the manuscript presented in an intelligible fashion and written in standard English?

Reviewer #1: Yes

Reviewer #2: Yes

6. Review Comments to the Author

Reviewer #1: Thank you for revising the manuscript. All my comments have been addressed. I believe it is now ready for publication.

Reviewer #2: 1. The main researching part of this article only briefly lists examples and characteristics of four categories, requiring more in-depth theoretical exploration and more innovative discoveries.

2. There are some viewpoints in the Discussion section of the article that lack supporting evidence and are rather casual, such as' fragmented structures of the current region '. There are still many conclusions that come from previous literature or subjective imagination, which cannot be directly derived from the analyzing results of this article, such as "the campaign style atmosphere for a specific policy right not last for an extended period. The principles and spirit of policy content may be downgraded to bureaucratic red tape".

7. PLOS authors have the option to publish the peer review history of their article (what does this mean?). If published, this will include your full peer review and any attached files.

Reviewer #1: No

Reviewer #2: No

---

## [Editor Report · Decision Letter 2]

2 Oct 2023

Motivational discourse and campaign-style mobilisation : A positive discourse analysis of language poverty alleviation discourse in China

PONE-D-22-35166R2

Dear Dr. Yu,

We’re pleased to inform you that your manuscript has been judged scientifically suitable for publication and will be formally accepted for publication once it meets all outstanding technical requirements.

Kind regards,

Bing Xue, Ph.D.

Academic Editor

PLOS ONE
---

## [Editor Report · Acceptance letter]

4 Oct 2023

PONE-D-22-35166R2 

Motivational discourse and campaign-style mobilisation: A positive discourse analysis of language poverty alleviation discourse in China 

Dear Dr. Yu:

I'm pleased to inform you that your manuscript has been deemed suitable for publication in PLOS ONE. Congratulations! Your manuscript is now with our production department. 

Kind regards, 

on behalf of

Professor Bing Xue 

Academic Editor

PLOS ONE